An electronic voting scheme based on homomorphic encryption and decentralization

http://orcid.org/0000-0002-9388-5690 Yuan Ke 1 2
Sang Peng 1
Zhang Suya 1
Chen Xi 3
Yang Wei 1 yangwei@henu.edu.cn
Jia Chunfu 1 4
1 Henan University, School of Computer and Information Engineering , Kaifeng, Henan , China
2 Henan University, Henan Province Engineering Research Center of Spatial Information Processing , Kaifeng, Henan , China
3 Henan University, International Education College , Zhengzhou, Henan , China
4 Nankai University, College of Cybersecurity , Tianjin , China
Akleylek Sedat
Electronic publication date: 2023 Oct 18
Publication date: 2023
Volume: 9
Electronic Location ID: e1649
Received 2023 May 23; Accepted 2023 Sep 21
Copyright: © 2023 Yuan et al.
Copyright year: 2023
Copyright holder: Yuan et al.
License: This is an open access article distributed under the terms of the Creative Commons Attribution License, which permits unrestricted use, distribution, reproduction and adaptation in any medium and for any purpose provided that it is properly attributed. For attribution, the original author(s), title, publication source (PeerJ Computer Science) and either DOI or URL of the article must be cited.
License URL: https://creativecommons.org/licenses/by/4.0/

Keywords: Electronic voting, Homomorphic encryption, Decentralization

Funding: National Natural Science Foundation of China 61972073, 61972215, 62066040 Natural Science Foundation of Tianjin 20JCZDJC00640 Key Specialized Research and Development Program of Henan Province 222102210062 Basic Higher Educational Key Scientific Research Program of Henan Province 22A413004 National Innovation Training Program of University Student 202110475072 This work was supported by the National Natural Science Foundation of China (61972073, 61972215, 62066040), the Natural Science Foundation of Tianjin (20JCZDJC00640), the Key Specialized Research and Development Program of Henan Province (222102210062), the Basic Higher Educational Key Scientific Research Program of Henan Province (22A413004), and the National Innovation Training Program of University Student (202110475072). The funders had no role in study design, data collection and analysis, decision to publish, or preparation of the manuscript.

==============================
Compared with paper-based voting, electronic voting not only has advantages in storage and transmission, but also can solve the security problems that exist in traditional voting. However, in practice, most electronic voting faces the risk of voting failure due to malicious voting by voters or ballot tampering by attackers. To solve this problem, this article proposes an electronic voting scheme based on homomorphic encryption and decentralization, which uses the Paillier homomorphic encryption method to ensure that the voting results are not leaked until the election is over. In addition, the scheme applies signatures and two layers of encryption to the ballots. First, the ballot is homomorphically encrypted using the homomorphic public key; then, the voter uses the private key to sign the ballot; and finally, the ballot is encrypted using the public key of the counting center. By signing the ballots and encrypting them in two layers, the security of the ballots in the transmission process and the establishment of the decentralized scheme are guaranteed. The security analysis shows that the proposed scheme can guarantee the completeness, verifiability, anonymity, and uniqueness of the electronic voting scheme. The performance analysis shows that the computational efficiency of the proposed scheme is improved by about 66.7% compared with the Fan et al. scheme (https://doi.org/10.1016/j.future.2019.10.016).

Introduction

The voting election is an important way to ensure the fairness of the election. The earliest voting election can even be traced back to the ancient Greek period. But the traditional voting scheme is time-consuming, has low turnout, and may appear to result in favoritism. With the advantages of high efficiency, convenience and rapidity, the electronic voting scheme has gradually received people’s attention in recent years.

E-voting scheme enjoy the convenience of information technology and necessarily bear its risks. Cable et al. (2023) details a series of key issues regarding voter registration security. Park et al. (2021) discusses the security risks that internet voting and blockchain voting may face in real-world applications. del Blanco, Alonso & Alonso (2018) summarizes the classic cryptographic schemes applied to remote electronic voting systems with their challenges.

The current stage of encrypted electronic voting schemes can be divided into hybrid network (Mix-net) based electronic voting schemes (Fujioka, Okamoto & Ohta, 1993; Chang et al., 2016; Islam et al., 2017; Alam et al., 2021; Haines, Goré & Sharma, 2021), blind signature-based electronic voting schemes (Kumar, Katti & Saxena, 2017; Aziz, 2019; Kumar, Chand & Katti, 2020; Carcia, Benslimane & Boutalbi, 2021), and homomorphic encryption-based electronic voting schemes (Gong et al., 2016; Yang et al., 2018; Fan et al., 2019; Anjima & Hari, 2019), depending on their implementation.

Mix-net was introduced by Chaum (1981) to enable anonymous communication and protect the privacy of voters. Mix-net is a routing protocol that enables hard-to-trace communication through intermediate nodes, where each node receives messages from multiple senders as a way to mix messages from multiple users and send the mixed messages to the next node. As a result, it is difficult for eavesdroppers to trace end-to-end communications. However, the scheme is unstable, and if any node fails, the entire voting activity is disrupted.

The concept of blind signature was first introduced by Chaum (1983). It is a two-party interaction protocol used to protect user privacy, where the voter first blinds the original message and sends it to a trusted third-party institution; the third-party institution signs and returns it to the voter without access to the message content; and finally, the user gets the correct signature of the third-party institution on the original message after removing the blinding. However, blind signature-based electronic voting schemes require anonymous channels to transmit ballots, but in reality, it is difficult to achieve fully anonymous channels. Therefore, blind signature-based electronic voting schemes are currently not widely used.

Decentralization is a characteristic of an organizational or systemic structure in which power, decision-making, and control are dispersed among multiple independent individuals or nodes, rather than being concentrated in a single central agency or entity. In a decentralized system, individual nodes can participate equally with each other and there is no single authority that can manipulate the entire system alone. This decentralized structure aims to increase the transparency, security, reliability and resistance to attack of the system. Through decentralization, decision-making and control are dispersed to multiple independent entities, eliminating the need for a single control by a central authority. Such a decentralized structure ensures that each step in the voting process can be independently scrutinized and validated, ensuring openness and transparency. Decentralized voting systems eliminate reliance on a single central authority, making the process more credible. The delegation of authority to multiple independent entities ensures that the voting process is more fair and impartial and not subject to manipulation by individual organizations or individuals.

Homomorphic encryption was originally proposed by Rivest, Adleman & Dertouzos (1978), and its basic idea is to integrate and compute the encrypted messages directly without decryption. When applied to electronic voting, the homomorphic encryption property makes the secrecy of the intermediate results of voting guaranteed and the trustworthiness of the final results of voting greatly improved. The electronic voting scheme based on homomorphic encryption has been widely used for its stability and security features. The scheme can be divided into the ElGamal scheme (Cramer, Gennaro & Schoenmakers, 1997; Azougaghe, Hedabou & Belkasmi, 2015; Jabbar & Alsaad, 2017; Wade & Gill, 2022), homomorphic scheme on integers (Aung et al., 2019), and the learning with errors over rings/learning with errors (LWE/RLWE) based scheme (Brakerski & Vaikuntanathan, 2014; Maringer, Puchinger & Wachter-Zeh, 2021) according to the different homomorphic encryption methods. However, most of the above schemes fail to achieve multi-candidate voting, and although the ElGamal scheme (Cramer, Gennaro & Schoenmakers, 1997) achieves multi-candidate voting, the ElGamal algorithm uses multiplication in both the process of homomorphic encryption and decryption of plaintexts, and in practical applications, the number of ballots is often huge, and the ElGamal algorithm requires a large number of multiplicative decentralized operations for the decryption and encryption process of ballots, thus computational inefficiency, which generates a not-so-subtle time overhead and sometimes even affects the conduct of the election. The LWE/RLWE (Brakerski & Vaikuntanathan, 2014; Behera & Prathuri, 2022) scheme achieves multi-candidate voting through multiple voting keys, which is equivalent to conducting multiple ballots, so the scheme operates inefficiently. In addition, the security of the key and the timed-release voting results are also problems that most homomorphic encryption-based electronic voting mechanisms fail to solve when applied in practice. To avoid the leakage of voting results before the end of the election, Braunlich & Grimm (2011) proposed an electronic voting scheme that combines the Paillier-based homomorphic encryption algorithm with the idea of gated key sharing. The scheme uses the Shamir secret sharing mechanism (Fujioka, Okamoto & Ohta, 1993), which divides the homomorphic private key into several fractions, and then distributes these fractions to different candidates as a guarantee that the election results will not be decrypted until a specified time is reached. However, the Shamir mechanism makes the scheme not as secure as an electronic voting scheme should be, and under the condition of sufficient arithmetic power, an attacker can reconstruct the slice into a complete private key by a finite number of operations, which will lead to the failure of the whole election. In 2018, Srivastava, Dwivedi & Singh (2018) proposed a decentralized voting scheme based on blockchain technology, and although the application of blockchain technology can provide security and transparency for electronic voting, there are still some drawbacks, such as the need for strong computing power and energy consumption, and blockchain technology requires a lot of computing power and energy to operate, which can lead to high operational costs and slower transaction speed. There is also the issue of anonymity. In an election, everyone should have the right to privacy in the election, but when using blockchain technology for voting, since all transaction records are public, it may lead to the identity of the voter being compromised, thus affecting the fairness of the election.

The Paillier encryption algorithm was proposed by Paillier (1999), and in the literature (López-García, Perez & Rodríguez-Henríquez, 2013), it is mentioned that although the Paillier public-key cryptosystem is too inefficient for its encryption and decryption, the homomorphism of Paillier consumes relatively less time in ciphertext operations, which is well suited to electronic voting in the counting process using the homomorphism of homomorphic encryption. The homomorphism of the Paillier encryption is very suitable for the vote accumulation in the counting process. Therefore, Paillier encryption is chosen as the main encryption algorithm in this article. When the homomorphic encryption mechanism works in the electronic voting scheme, the ballots are sent to the counting center after homomorphic encryption, and the voting results are uniformly disclosed after the election. However, in the intermediate process, the voting system is vulnerable to attacks and there is no guarantee that the homomorphic private key used for decryption can be known until the end. Once the homomorphic private key is known to the wrong person in advance, the election result will become unreliable and the whole election will fail. To prevent this phenomenon, we introduce the public-private key pair of the cloud-based counting center and the public-private key pair of the voter in the homomorphic encryption-based electronic voting scheme, and the private keys of the counting center and homomorphic decryption are held by different candidates, and the security of the ballot results is guaranteed because of the competition among different candidates.

In order to ensure the accuracy and fairness of the voting results, while improving the efficiency and saving the cost of calculation, this article proposes an electronic voting scheme based on homomorphic encryption and decentralization (HED-Voting) to solve the problem of voting failure due to malicious voting by voters or ballot tampering by attackers. The main contributions of this article are as follows. (1) We propose a new voting scheme that innovatively introduces decentralization in the management of the secret key used for decryption, distributes encrypted ballots with homomorphic decryption keys to different candidates with opposing relationships to ensure that no party knows the voting results in advance during the election process. It effectively counteracts the security risks of private key leakage and internal corruption.

(2) We propose a new encryption scheme that uses a homomorphic encryption mechanism to ensure self-counting and avoid the delay caused by the decryption of the ballot results after the voting is completed. RSA is adopted for the ballot to sign the user’s ballot, and the signed ballot is encrypted in two layers using the Paillier homomorphic encryption mechanism and the ElGamal asymmetric encryption mechanism to ensure that the power-sharing condition can be established.

This article is organized as follows: “Preliminaries” provides the pre-requisite knowledge needed in this article, including attacker model, Paillier homomorphic encryption mechanism, Elgamal public key encryption mechanism and basic terminology. “E-Voting Scheme Based on Homomorphic Encryption and Decentralization” introduces the detail scheme of the HED-Voting. “Security Analysis” analyzes the security aspects of the e-voting scheme proposed in this article. “Performance Analysis” analyzes the operational efficiency of the HED-Voting scheme. “Conclusion” provides a concluding outlook for entire article.

Preliminaries

In this subsection, we focus on explaining the pre-requisite knowledge needed in this article.

The basics

In this subsection, we will give the definitions of semi-honest attacker model, malicious attacker model, Paillier homomorphic encryption mechanism, and ElGamal public key encryption mechanism used in this article.

Attacker model

The attacker model can be divided into the semi-honest attacker model and the malicious attacker model.

Definition 1 Semi-honest attacker model. The semi-honest attacker model means that the attacker will reach an agreement with the participant through bribes and threats, which stipulates that the participant will not quit before the end of the activity, and the participant can honestly carry out sending their calculation results during the activity, i.e., they guarantee that they will not tamper with the final results of the activity, but the participant must inform the attacker of all relevant information about the whole activity. This includes historical communication information, calculation results, etc.

Definition 2 Malicious attacker model. The malicious attacker model means that the participant who has made an agreement with the attacker will no longer participate in the activity honestly and will send false results during the activity in order to achieve the purpose of tampering with the results of the activity.

Paillier homomorphic encryption mechanism

The Paillier homomorphic encryption (PHE) is a typical asymmetric homomorphic encryption algorithm, where the key used for encryption (public key) and the key used for decryption (private key) are different. Unlike symmetric encryption algorithms, this encryption algorithm does not require sharing the key between the sender and the receiver and has better security.

Definition 3 Paillier homomorphic encryption. The Paillier homomorphic encryption mechanism consists of algorithm 3-tuple EPHE={GenPHE,EncPHE,DecPHE}, where

GenPHE: Given two random large prime numbers p and q, n=pq, λ=lcm(p−1,q−1). Take g∈Zn2∗, and satisfy gcd(L(gλmodn2),n), here: L(x)=x−1n. Get the encryption public key as PK=(n,g) and the private key as SK=(p,q).

EncPHE: The data is encrypted with a randomly selected integer r∈Zn∗, the plaintext m∈Zn∗, and the encryption result is c=gmrnmodn2. Where: c is the ciphertext data corresponding to the plaintext m and c∈Zn2∗. It can be seen that for the same ciphertext m, the randomly selected values of r in the encryption process may be different, and the corresponding ciphertext data after encryption may not be the same, thus ensuring the semantic security of the ciphertext data.

DecPHE: The data is decrypted, m=D(c)=L(cλmod(n2)L(gλmod(n2)modn.

ElGamal public key encryption mechanism

ElGamal Public key encryption (EPKE) is an internationally recognized public key cryptosystem, its encryption algorithm is based on the Diffile–Hellman key exchange algorithm, which was proposed by Taher ElGamal in 1985, its security is based on the problem of computing discrete logarithms over a finite domain, compared to RSA algorithm, ElGamal algorithm can resist replay attacks.

Definition 4 ElGamal Public key encryption. The ElGamal Public key encryption mechanism consists of algorithm 3-tuple EEPKE={GenEPKE,EncEPKE,DecEPKE}, where

GenEPKE: The system randomly selects a large prime number p ( p must satisfy the existence of q, and p|q−1), so that g is an original root of the multiplicative group Zp∗. Generate a random integer x ( 1≤x≤p−1) and compute h=gxmodp. Use ( p,q,g) as the user’s public key and x as the user’s private key.

EncEPKE: The data is encrypted. Let the plaintext that the user wants to encrypt be m, and m<p, randomly selected integer k(1≤k≤p−1), and calculated ciphertext c=<a,b>=<gkmodp,m⋅hkmodp>.

DecEPKE: The user decrypts the ciphertext with the private key, and get the plaintext m=b⋅(ax)−1modp.

Key notations

For presentation convenience, we list the key notations used in our work in Table 1.

Table 1 Key notations used in this article.

Symbol	Description	
e-voting	Electronic voting	
CC	Counting center	
CA	Certificate authority	
KC	Key center	
PC	Publicity center	
HEpub	The public key of homomorphic encryption	
HEpriv	The private key of homomorphic encryption	
CCpub	The public key of counting center	
CCpriv	The private key of counting center	
ui	User i who is voter	
upub	The public key that belongs only to ui.	
upriv	The private key that belongs only to ui.	
mi	The plaintext of the vote that belongs to ui.	
ci′	The first layer of ciphertext after using HEpub homeomorphic encryption	
s	Generated by using upriv, which together with ci′ forms the second layer of ciphertext	
C	Generated by using CCpub to encrypt s	
c′	The counting result in ciphertext	
m	The counting result in plaintext	
Coste	The time required for a decentralized multiplication operation	
Nv	The number of voters	
Nc	The number of candidates	
Costvoter	For each voter client, the time cost of the voting phase	
CostCC	The time cost for processing a ballot in the CC	

E-voting scheme based on homomorphic encryption and decentralization

A model based on a homomorphic encrypted e-voting scheme in a decentralized environment is given along with a specific scheme.

HED-voting model

The goal of the model is to use the Paillier homomorphic encryption enabling specific computational operations even when the data is encrypted. This enables data to be processed and computed without exposing the plaintext, effectively preserving data privacy, method to ensure that the ballot results are not compromised until the election is over. By signing the ballot and encrypting it in two layers, the security of the ballot during transmission and the establishment of the decentralized scheme is guaranteed. The decentralized scheme is established by distributing decision-making and control among different independent entities through the allocation of dual-layer encrypted keys to different candidates.

Suppose in a real-life election campaign, the candidates have a competitive relationship with each other, and two candidates with opposing relationships are randomly selected and given different privileges, one of them is responsible for the management of the key, and the other is responsible for the management of the cloud-based CC, with no interference between them. After the ballots are encrypted and signed, they are sent directly to the tally center for verification and homomorphic operation, and when the voting is over, the results are sent to the public center, and then the results are decrypted and publicized using the homomorphic private key. Since both the organization in charge of managing the KC and the organization in charge of the cloud-based CC lack the conditions for decryption, neither of them can obtain the voting results before the election ends. A schematic diagram of the HED-Voting is shown in Fig. 1.

Figure 1 E-voting system based on homomorphic encryption and decentralization scheme.

Definition 5 HED-Voting system. The HED-Voting system consists of Voter, Counting Center, Certificate Authority, Key Center, Publicity Center, and algorithm 10-tuple

EHED-Voting={Setup_HE,Setup_CC,User_KeyGen,Enc_HE,Rsa_Sig,Enc_CC, Dec_CC,Rsa_Ver,Count_CC,Dec_HE}, where

Setup_HE → (HEpub,HEpriv). Generate the public-private key pair for the KC. Enter the security parameters to generate the public-private key pair for homomorphic encryption at (HEpub,HEpriv).

Setup_CC → CCpub. Initialize and generate the CC public-private key pair (CCpub,CCpriv) and output the CC public key CCpub.

User_KeyGen → (upub,upriv). Generate the user’s public-private key pair (upub,upriv).

Enc_HE → ci′. The ballot is homomorphically encrypted for each voter. The ballot mi is encrypted using HEpub to obtain ci′.

Rsa_Sig → (ci′,s). Use upriv to sign ci′ to obtain (ci′,s).

Enc_CC → C. Use CCpub to encrypt to obtain C.

Dec_CC → (ci′,s). Decrypt C using to obtain (ci′,s).

Rsa_Ver → ci′. Use upub to verify and get ci′ when it passes.

Count_CC → c′. Update c′ by homomorphic the ciphertext ci′ with the previously counted votes c′.

Dec_HE → m. Use HEpriv to decrypt the plaintext m corresponding to the ciphertext c′.

The behavior of the HED-Voting system includes the following four stages. And the encryption, decryption, and counting of ballots during the voting process is shown in Fig. 2. (1) Initialization (corresponding algorithms Setup_HE and Setup_CC): First, initialize the KC and CC to generate homomorphic encryption public-private key pairs and CC public-private key pairs. The generated homomorphic encryption private key and CC private key are sent to two candidates with direct competition. Due to the direct conflict of interest, the two candidates will not conspire to exchange the private keys held by both parties. Due to the double-layer encryption used by the scheme, the private key held by one candidate alone is not enough to decrypt the voting information in advance.

(2) Authentication (corresponding algorithm User_KeyGen): After initialization, users are registered. Users are required to register locally in the corresponding software and bring their personal information to the CA to obtain a public-private key pair and a trusted certificate after identity verification. Before the voting process, the CA sends the user’s certificate to the CC.

(3) Voting phase (corresponding algorithm Enc_HE, Rsa_Sig, and Enc_CC, Dec_CC, and Rsa_Ver and Count_CC): In the voting stage, the voter signs and encrypts the ballot using HEpub, upriv and CCpub in turn, and then sends the ballot ciphertext and upub to the CC. After the ballot reaches the CC, upub is tokenized and the encrypted ballot is stored in the CC′s database. After receiving the ballot, the CC decrypts the ballot with CCpriv and then verifies the signature with the corresponding voter’s public key, after which the ciphertext is homomorphic with the previous voting result and the voting result is updated.

(4) Result Announcement (corresponding algorithm Dec_HE): After the voting is finished, the final voting results are sent to the PC. The organization in charge of managing KC decrypts the ciphertext.

Figure 2 The encryption, decryption, and counting of ballots during the voting process.

HED-voting solution

The concrete construction scheme of HED-Voting includes the following 10 algorithms.

Setup_HE. For the KC, randomly select two large prime numbers p and q that satisfy gcd(pq,(p−1)(q−1))=1. For p and q, perform the following operations.

① Calculate n1=pq and λ=lcm(p−1,q−1).

② Choose a random large integer g∈Zn12∗ and calculate μ=[L(gλmodn12)]−1modn1. Where L(x)=x−1n1.

Obtain the public key HEpub=(n1,g) and private key HEpriv=(λ,μ) of the homomorphic encryption.

Setup_CC. For the CC, perform the following operations.

① A randomly selected generating element g1 is used to generate a q1 order cyclic group G using g1.

② Randomly select x, and x from G, satisfy 1<x<q1−1.

③ Calculate h=g1xmodq1.

Obtain the public key CCpub=(q1,g1,h) and private key CCpriv=x of the CC.

User_KeyGen. Obtain the user public key upub=(n2,e) and the user private key upriv=d, and perform the following operations.

① Select two large prime numbers p2 and q2, and calculate the product n2=p2q2.

② Compute φ(n2)=(p2−1)⋅(q2−1), the security of this step is based on Little Fermat’s theorem.

③ Randomly select e: 0<e<φ(n2), and gcd(e,φ(n2))=1.

④ d=e−1modφ(n2). This gives the public-private key pair upub=(n2,e), upriv=(p2,q2,d).

Enc_HE. For each voter’s ballot mi, encrypted with the homomorphic encryption public key HEpub, perform the following operations.

① Mapping mi to 0≤x<n1 where n1=pq.

② Select r∈Zn1∗ while satisfying (r,n1)=1, i.e., gcd(r,n1)=1.

③ Calculate the ciphertext at ci′=gmr1nmodn12.

Rsa_Sig: This step uses the voter’s private key upriv to generate a digital signature s=sigupriv(ci′)≡(ci′)dmodn2. This gets ct_i=(ci′,s).

Enc_CC. Here the ciphertext with the signature ct_i=(ci′,s) is encrypted again with the public key CCpub. Perform the following operations.

① Randomly select y, and y satisfy 1<y<q1−1.

② Calculate a=g1ymodq1,s1=hymodq1.

③ Maps ct_i to an element ct_i′ on G.

④ Calculate b=ct_i′s1.

Get the ciphertext C=<a,b> and send C to the CC.

Dec_CC. The CC decrypts the ciphertext C with the private key CCpriv, perform the following operations.

① Calculate b(a−1)x=ct_i′hyg1xy(g1xy)−1=ct_i′.

② Remap ct_i′ to ct_i=(ci′,s).

Rsa_Ver. Use the voter’s public key upub, verupub(ci′,s),ci′′≡semodn2. If

ci′′≡ci′modn2, then the signature is valid;

ci′′≠ci′modn2, then the signature is invalid.

Count_CC. Update c′ by homomorphing the ciphertext ci′ with the previously counted votes c′. For the ciphertexts ci′,c′∈Zn12∗, whose corresponding plaintexts are mi,m∈Zn and ri,r∈Zn∗, we have ci′=E(mi,ri),c′=E(m,r), yielding

ci′⋅c′=E(mi,ri)⋅E(m,r)=gmi+m⋅(ri⋅r)n1modn12=E((mi+m),(ri⋅r)).

After update, c′=gmi+m⋅(ri⋅r)n1modn12=E((mi+m),(ri⋅r)).

Dec_HE. Use HEpriv to decrypt the ciphertext c′. For the ciphertext c′∈Zn12∗, compute the ciphertext m: The ciphertext will be decrypted.

m=D(c′)=L((c′)λmodn12)L(gλmodn12)modn1=L((c′)λmodn12)μmodn1.

Security analysis

In this subsection, we will analyze the security aspects of the e-voting scheme proposed in this article.

The security model of this scheme is given first. In this scheme, it is assumed that the selected CA is “honest and curious”: the CA will perform its duties according to the election rules and will not allow malicious attackers to pass the certification, and at the same time, it will presume potential semi-malicious attackers based on its own analysis, so as to prevent illegitimate voters from leaking election and interfere with the results of the election. The different candidates involved in the election are in an adversarial relationship, and there is no cooperation between them. In addition, we believe that the CC is absolutely secure during the election campaign.

The following is a security analysis as well as a semantic security analysis for the potential threats that may be encountered in the e-voting scheme proposed in this article to demonstrate the security of the scheme in the relevant application scenarios.

Potential threat security analysis

Theorem 1. Only certified voters are allowed to vote.

Proof. In the registration stage, voters must present their personal identification information to the trusted registration center and successfully authenticate before they can obtain their personal public-private key pair. Not only that, only after the certification center successfully verifies the authenticity of personal information will they obtain their personal certification, otherwise, the public-private key pair applied for in the registration stage is invalid. After successful authentication, the certification center synchronizes the voter’s public key to the CC. In the voting stage, voters need to select their target candidates in addition to entering their public keys. If the voter’s identity is illegal, his or her public key is invalid and he or she cannot participate in the voting. The theorem is proved.

Theorem 2. In this scheme, the results of election voting are fair and the probability of early leakage of results is extremely small, ensuring the completeness of the e-voting scheme.

Proof. In this scheme, CCpriv and HEpriv are held by different entities. Since there is an adversarial relationship between different candidates and there is no collusion between them, for the candidate holding the private key, cannot obtain the ballot without obtaining CCpriv and thus cannot know the voting result; for the CC, since it does not obtain the homomorphic encryption private key and the probability of breaking the HEpriv is negligible, he cannot know the voting. The result of the voting is not known to the candidate who holds the homomorphic encryption private key. The theorem is proved.

Theorem 3. After the polls close, each voter can check whether his or her ballot has been tampered with, ensuring the verifiability of the voting scheme.

Proof. After the polls close, the PC will publish all the encrypted ballots and the CCpriv at the same time as the PC publishes the voting results. Each voter can find his or her ballot C=<a,b> according to his or her public key, and the voter can check whether his or her ballot has been tampered with before it reaches the CC by decrypting the ciphertext according to the CCpriv to get ct_i=(ci′,s). The theorem is proved.

Theorem 4. The e-voting scheme implemented in this scheme does not reveal the privacy of the voters and guarantees the anonymity of the e-scheme.

Proof. In the registration stage, this solution uses local software registration, and users do not need to go to the registration center to register. In the voting stage, the user does not need to present the identity verification information involving personal privacy, but only the public key obtained in the registration stage, so no personal information will be disclosed. The theorem is proved.

Theorem 5. In the election process, each voter can only vote once, which guarantees the uniqueness of the voting scheme.

Proof. We construct a database in the CC in which the fields consist of the voter’s encrypted ballot c′, the voter’s public key upub, and a flag bit. In the voting phase, voters select their target candidates and enter their public keys. When the voter’s ballot and the public key are sent to the CC if the public key is used for the first time, the voter’s ballot is stored in the database and then the flag is set to 1. If the public key is not used for the first time, the data is not stored in the database and does not participate in the homomorphic operation. The theorem is proved.

Theorem 6. The voter behind the ballot cannot repudiate his or her vote, and each voter is one-to-one and bound to the ballot he or she casts, which is difficult to tamper with by others.

Proof. Voters need to use their own signature key to sign the ballot they cast in the voting stage, and the ballot is allowed to enter the encryption and calculation process only after it is signed by the voter behind it. This guarantees that the voter’s ballot is difficult to be tampered with by others and that the voter cannot deny his or her vote, i.e., it guarantees the immutability and non-repudiation of the ballot.

Table 2 compares the characteristics of our HED-voting scheme and the HSE-voting scheme of Fan et al. (2020). Compared with the HSE scheme, in our scheme, the voter behind the ballot cannot deny the ballot he or she casts, each voter is one-to-one and bound to the ballot he or she casts, and the ballot is difficult to be tampered with by others, which guarantees the immutability and the non-repudiation of the scheme.

Table 2 Comparison of characteristics of e-voting schemes.

Characteristics	HSE	HED	
Eligibility	✓	✓	
Uniqueness	✓	✓	
Privacy	✓	✓	
Verifiability	✓	✓	
Immutability	✗	✓	
Non-repudiation	✗	✓	

Semantic security analysis

We formalize here the analysis of the semantic security of cryptographic regimes by defining security games. In the security game, we statute the security of the regime to the security of Paillier encryption and Elgamal encryption. We define the security game G0 simulate the regime in a real scenario, where the adversary A simulates an eavesdropper whose aim is to decipher the ciphertext, the challenger C simulates a user who honestly executes the protocol. We treat the hash function in the security game as a random prediction machine, and the challenger C needs to interrogate the random prophecy machine when computing the hash value. For each query, the prophecy machine will return a uniformly random output. For recurring queries, the prophecy machine will return the same result. The security game G0 is described in detail as follows:

Initialization phase: Challenger C generates homomorphic encrypted public-private key pairs (HEpub,HEpriv), random number rP∈Zp∗, rE∈Zn∗, CC public-private key pair (CCpub,CCpriv), RSA signature public-private key pair (upub,upriv).

Challenge phase: A outputs two equal-length challenge plaintexts (M0,M1). C randomly chooses bεR{0,1}, calculates ci′=gMbrPn(modn2), and C=(grE1,ci′s1). Finally, C returns ci′ and C to A.

Speculation phase: Adversary A outputs b′ If b=b′, then A wins G0. The probability advantage of having in is defined as Adν(A)=|Pr[b=b′]−12|.

Definition 6. If for any PPT adversary A, there exists a negligible value ε that satisfies Adν(A)≤ ε, then the eavesdropper in this scheme is considered to be unable to break the semantic security of the homomorphic encryption regime.

Theorem 7. If the Paillier encryption, Elgamal encryption used in the scheme is semantically secure, the eavesdropper in this scheme cannot break the semantic security of the encryption regime under the random prediction machine model.

Proof. Theorem 1 is proved here by defining multiple indistinguishable security games.

G0I: The game is the same as G0.

G1I: The game is similar to G0I, except that the challenger C replaces ci′ replaced by a random number of equal length R0. Since Paillier encryption is semantically secure, the private key in HEpriv to the adversary A in the case of secrecy G1I with G0I has indistinguishability.

G2I: The game is similar to G1I, except that the challenger C replaces C with a random binary of the same format R1. Since Elgamal encryption is semantically secure, the private key in CCpriv to the adversary A in the case of secrecy G2I with G1I has indistinguishability.

In G2I, adversary A receives the messages of (R0,R1), all of which are uniformly random values, independent of the challenge plaintexts (M0,M1). Therefore, A in the speculation phase only random output b′, its probability advantage is negligible, i.e., Adν(A)=|Pr[b=b′]−12|≤ε. Since G2I and G0 are indistinguishable, while G0, A and (M0,M1) simulate the present system, the eavesdropper and the plaintext, respectively, it can be assumed that the eavesdropper cannot break the semantic security of the present encryption regime. The theorem is proved.

In summary, this scheme can guarantee the completeness, verifiability, anonymity, and uniqueness of the e-voting scheme, and only authenticated voters can vote, preventing illegitimate voters from leaking election information and interfering with the election results. Meanwhile, eavesdroppers cannot break the semantic security of this encryption scheme.

Attack model analysis

The scheme proposed in this article can effectively resist the following attacks.

Spoofing attack

In the scheme designed in this article, the voter’s real identity ID and ballot ciphertext are unlikely to be leaked, so that the attacker can not obtain the voter’s identity information from it, so as to forge the identity of legitimate voters.

Man-in-the-middle attack

The scheme designed in this article is mainly based on the composite residuosity problem, no matter the voter submits successfully or not, the malicious attacker can not crack the ballot ciphertext.

Denial of service attack

In the scheme designed in this article, the system does not receive ballots without trusted certificates, and for each voter only has the right to submit a valid ballot once, it is difficult to occur serious denial of service attack symptoms. After the ballot is submitted to the system, it will be homomorphic computed in ciphertext state directly, even if some denial-of-service attack occurs, it will not affect the security and privacy of the data.

Performance analysis

In this subsection, we analyze the operational efficiency of the HED-Voting scheme. Since the time cost consumed by the add and multiply operations is negligible compared to the time cost required by the multiply decentralized operation, we choose to use the time consumed by the multiply decentralized operation in the voting scheme as a measure of the performance of the voting scheme.

All test behaviors are performed in the following hardware environment: CPU, 12th Gen Intel(R) Core(TM) i7-12700H; Memory, 16 GB, 4,800 MHZ DDR5.

Testing with the gmpy2 python module (https://gmpy2.readthedocs.io/en/latest/) gives Coste=0.0068s. To provide a clearer count of the time cost of the voting scheme, we aggregate the time cost for each of the different institutions in the entire election campaign, but ignore the time required by the PC since it only performs a decentralized multiplication operation.

Performance analysis of the voter client

In the voting stage, each voter needs to sign the ballot once and encrypt it twice: first, the ballot is encrypted with the HEpub, and the homomorphic encryption algorithm chosen in this scheme is the Paillier encryption algorithm, which requires one multiplication decentralized operation during encryption; then, the voter signs the ballot with his or her own private key, and we choose the RSA signature algorithm to sign the ballot. Finally, the ballot is encrypted with the CCpub, and the Elgamal algorithm is used to encrypt the ballot, which requires two decentralized multiplication operations. Therefore, for each voter client, the time cost of the voting phase.

COSTvoter=4×Coste×Nc.

Performance analysis of the CC

In the election process, the CC needs to decrypt the encrypted ballot and verify the signature. In addition, the CC also needs to perform homomorphic addition on the ciphertext, but since homomorphic addition is essentially a multiplication of large integers, the time consumed is negligible compared to the decentralized multiplication operation. After analysis, we know that the CC performs one multiplication decentralized operation for both decryption and signature verification of ballots, so the time cost for processing a ballot in the CC is

COSTCC=2×Coste×Nc

We assume that each voter only needs to cast one vote for his or her target candidate during the voting process and does not need to perform any operation on the other candidates, and when the votes are counted, the number of votes for the selected candidate increases by one and the number of votes for the other candidates who are not selected remains the same. Based on this assumption, we compared the time cost required for the proposed HED-Voting scheme and the HSE-Voting scheme (Fan et al., 2020), as shown in Table 3.

Table 3 Comparison of calculated costs between HED-Voting and HSE-Voting.

Programs	Calculated cost per voter	Counting cost of the CC	
HED-Voting	4×Coste×Nc	2×Coste×Nc×Nv	
HSE-Voting	9×Coste×Nc	6×Coste×Nc×Nv+4×Coste	

We ran five scenario simulations based on different numbers of voters ( Nv= 1,000, 2,000, 4,000, 7,000, 10,000) and counted the time spent by the CC to process all ballots. In the simulations, we assume that the number of candidates is five, That is, Nc=5 and that each voter also needs to vote for only one candidate. The results are shown in Fig. 3.

Figure 3 The time cost comparison for CC of the two schemes in the case of five candidates, the number of voters is respectively 1,000, 2,000, 4,000, 7,000, 10,000 ( Coste=0.0068s).

As can be seen from Fig. 3, the actual time costs required for both scenarios are consistent with the theoretical estimates of the time costs in Table 3. In all five scenario simulations, the computational cost of the HED-Voting scheme is lower than that of the HSE-Voting scheme, and the computational efficiency of the HED-Voting scheme is improved by about 66.7% compared with that of the HSE-Voting scheme. The scheme has the capability to handle a large number of voters and counting within an acceptable time frame.

HED-voting scheme computational complexity

For computational complexity analysis, we chose 512 bits for large primes and 1,024 bits for n. Modulo power and modulo inverse operations are performed using Python’s gmpy2 module, and Paillier key generation is performed using Python’s phe module. The program is executed on an Intel(R) Core (TM) i5-10210U CPU @ 1.60 GHz 2.11 GHz processor, 64-bit host, 16 GB RAM, and Python version 3.9.8. After executing the program, for a modulo multiplication operation Mul on the program, the operation time is about 0.0035 ms. To make the computational results independent of computer performance, the computational cost of the relevant basic operations in the program is calculated using the time consumption of Mul1 as the basic weight, as shown in Table 4.

Table 4 Relative time consumption of related operations.

Basic arithmetic operation	Identifier	Relative time consumption	
Modulo multiplications operation on Zn12∗	Mul1	1	
Modulo multiplication operation on Zn1∗	Mul2	0.33	
Modulo power operation on Zn12∗	Pow	715.41	
Modulo inverse operation on Zn12∗	Inv	77.43	
Pallier key generation	Key	10,801.63	

Based on the relative elapsed time of the relevant base operations in Table 4, the computational complexity of the following algorithms is calculated (some of them are no longer given because their elapsed time is negligible due to their short elapsed time), as in Table 5, For Setup_HE, it needs a Paillier key generation operation, so its base operation is Key. For Setup_CC, it needs a modulo power operation, so the base operation is Mul2. For User_KeyGen, a simulation operation is required, so the base operation is Inv. For Enc_HE, there is a modulo multiplication operation and a modulo power operation. Therefore, the basic operation is Pow+Mul1. For Enc_CC, there are two modulo power operations, so the base operation is 2Pow. For Dec_CC, there a For Rsa_Sig, there is one modulo power operation, so the base operation is Pow.re 3 modulo multiplications, three modulo powers, and one simulation operation, so the base operation is 3Mul2+3Pow+Inv. For Rsa_Ver, there is one modulo power operation, so the base operation is Pow. For Count_CC, there are two modulo operations and two modulo operations, so the base operation is 2Pow+2Mul1. For Dec_HE, there is one modulo operation and one modulo operation, so the base operation is Pow+Mul2.

Table 5 Computational complexity of the HED-voting scheme.

Algorithms	Basic operation	Complexity of calculation	
Setup_HE	Key	10,801.63	
Setup_CC	Mul2	0.33	
User_KeyGen	Inv	77.43	
Enc_HE	Pow+Mul1	716.41	
Rsa_Sig	Pow	715.41	
Enc_CC	2Pow	1,430.82	
Dec_CC	3Mul2+3Pow+Inv	2,224.65	
Rsa_Ver	Pow	715.41	
Count_CC	2Pow+2Mul1	1,432.82	
Dec_HE	Pow+Mul2	715.74	

Conclusion

In order to solve the problem of voting failure due to malicious voting by voters or tampering by attackers, this article proposes an e-voting scheme based on homomorphic encryption and decentralization. In this scheme, the use of the Paillier encryption algorithm and decentralization scheme ensures the security of voting results during the election process, and the signature and two-layer encryption of ballots avoid the tampering of ballots during the transmission process. Under the security model proposed in this scheme, the scheme also satisfies completeness, anonymity, self-counting, uniqueness, and supports multi-candidate election and ballot inspection, where only certified voters can vote, preventing illegitimate voters from leaking election information and interfering with the election results. The performance analysis of HED-Voting and HSE-Voting through the simulation of real scenarios shows that this solution is more efficient.

However, during the voting phase, all the voters’ actions are done online, and once the voters’ hosts are maliciously attacked and a botnet is formed, the election will face defeat. Therefore, designing a voting scheme that can avoid malicious attacks on software is our next research work.

Supplemental Information

Supplemental Information 1 The time cost comparison of 5 candidates for CC.

The x-axis coordinates indicate the number of voters and the y-axis indicates the cost of time required by the counting center. Those with a triangular pattern on top of the line segment represent the HED-Voting scheme, and those with a square pattern on top of the line segment represent the HSE-Voting scheme.

Click here for additional data file.

Supplemental Information 2 Time calculation of one multiplication power operation.

Click here for additional data file.

Supplemental Information 3 The number of voters.

Click here for additional data file.

Additional Information and Declarations

Competing Interests

Author Contributions

Data Availability

The authors declare that they have no competing interests.

Ke Yuan conceived and designed the experiments, analyzed the data, authored or reviewed drafts of the article, and approved the final draft.

Peng Sang conceived and designed the experiments, performed the experiments, analyzed the data, performed the computation work, prepared figures and/or tables, authored or reviewed drafts of the article, and approved the final draft.

Suya Zhang conceived and designed the experiments, performed the experiments, analyzed the data, performed the computation work, prepared figures and/or tables, authored or reviewed drafts of the article, and approved the final draft.

Xi Chen performed the experiments, analyzed the data, performed the computation work, prepared figures and/or tables, authored or reviewed drafts of the article, and approved the final draft.

Wei Yang analyzed the data, authored or reviewed drafts of the article, and approved the final draft.

Chunfu Jia analyzed the data, authored or reviewed drafts of the article, and approved the final draft.

The following information was supplied regarding data availability:

The code and raw data are available in the Supplemental Files, GitHub, and Zenodo:

- https://github.com/breeze-666/An-electronic-voting-scheme-based-on-homomorphic-encryption-and-decentralization

- Suya Zhang, & Peng Sang. (2023). An-electronic-voting-scheme-based-on-homomorphic-encryption-and-decentralization [Data set]. Zenodo. https://doi.org/10.5281/zenodo.8185828.

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
