# Peer review of "An electronic voting scheme based on homomorphic encryption and decentralization"

_PeerJ Computer Science, doi:10.7717/peerj-cs.1649_

## Round 0.1 · original submission · Major Revisions

The paper needs more effort before publication. The comments have been given.

·

Basic reporting

This direction holds significant promise in addressing key challenges in electronic voting systems. To further strengthen the research, the following aspects can be considered:

Elaborate on the scheme architecture and design: Provide a detailed explanation of the proposed electronic voting scheme based on homomorphic encryption and decentralization. Describe the components, interactions, and underlying mechanisms that ensure the security and integrity of the voting process.

Discuss the integration of homomorphic encryption: Explore how homomorphic encryption can be effectively integrated into the electronic voting scheme. Highlight the advantages of homomorphic encryption in terms of preserving privacy while enabling computations on encrypted data, thus allowing for secure vote counting and verification.

Address decentralization and transparency: Emphasize the importance of decentralization in the electronic voting scheme. Discuss how the decentralized nature of the system enhances transparency, resilience against attacks, and trust in the voting process. Consider the role of distributed ledger technologies, such as blockchain, in achieving decentralization and tamper-proof recording of votes.

Evaluate security and privacy guarantees: Conduct a thorough analysis of the security and privacy guarantees provided by the proposed scheme. Discuss potential vulnerabilities and countermeasures to mitigate threats, such as coercion or vote manipulation. Evaluate the resilience of the scheme against attacks and potential risks to voter privacy.

Consider usability and accessibility: Discuss the usability and accessibility aspects of the electronic voting scheme. Consider the user experience, ease of use, and inclusiveness of the system to accommodate diverse voters, including those with disabilities or limited technical expertise.

Conduct performance evaluations: Perform comprehensive performance evaluations to assess the efficiency and scalability of the proposed scheme. Consider factors such as computational overhead, communication costs, and the ability to handle a large number of voters and vote tallies within acceptable timeframes.

By addressing these aspects, the research can contribute to the development of a robust and secure electronic voting scheme based on homomorphic encryption and decentralization. This will enable trustworthy and verifiable voting processes while ensuring the confidentiality of votes and protecting against malicious activities.

Experimental design

see Basic reporting

Validity of the findings

see Basic reporting

Additional comments

see Basic reporting

Reviewer 2 ·

Basic reporting

The authors have addressed the reviewers' comments and concerns in their revised manuscript. The paper is well written, but I have a few comments/suggestions for the quality of the paper. References are not enough. Adding current studies on e-voting will be better for the quality of the paper. Tables and Figure 1 and Figure 2 are readable and appropriate. In line 391, figure 3 might be more precise. The authors have provided experimental code.

Experimental design

The main contribution of the paper is explained in line 111. The novelty of the paper is good, but it is not emphasized enough, and the results obtained in comparison with other architecture can be seen; it would also be better to emphasize that it is a refinement.

Validity of the findings

Security analysis is also given semantically. It would be better for the quality of the paper to include computational and communication complexity as complexity in the performance analysis section.

Cite this review as

Reviewer 3 ·

Basic reporting

This paper proposes an election voting system based on homomorphic encryption and digital signatures. The system has been analyzed for security but still has many questions to be answered.

- It would be better to cover it in a subsection to highlight the contributions.
- Algorithms in section 3.2 must be specified in the algorithm layout in Latex.
- One of the major disadvantages of homomorphic encryption is that it is computationally slow and requires a lot of resources. From this point of view, can it bring additional problems in critical tasks such as elections? This issue should also be discussed because homomorphic encryption can cause performance problems in real-time systems.
- The paper assumes that systems such as CC, CA, KC, PC are secure. However, relying on so many actors can also bring vulnerabilities. Therefore, trust in these systems should be questionable and auditable for a schema that is a potential attack target such as election. Detailed information on this is needed.
- The schema is analyzed for some security situations. However, it needs to be analyzed in terms of all the basic principles of information security (ex. integrity).
- The literature contains many blockchain-based schemes related to e-voting. In terms of security and performance, the difference and superiority of the proposed scheme from blockchain-based e-selection systems should be discussed.
- The e-voting schema should be more clearly represented by a sequence diagram that includes the sequences of actions.
- A detailed discussion section should be added and the conclusion section should be expanded.

Experimental design

- The works stated to be carried out for experimental design and performance analysis are explained rather superficially. It should be detailed. Which metrics, which methods were used?

Validity of the findings

- Security analysis is handled rather poorly. There is no analysis on how the system will defend against cyber attacks (DoS, MITM, impersonate etc.).
- In the analysis section, the e-voting scheme is compared with only one method. This table should be expanded to include other methods.

Cite this review as

---

## Round 0.2 · Major Revisions

The review process is now complete. While finding your paper interesting and worthy of publication, the referees and I feel that more work could be done before the paper is published. My decision is therefore to provisionally accept your paper subject to major revisions.

·

Basic reporting

Authors updated the paper as per my comments.

Experimental design

as above

Validity of the findings

as above

Additional comments

as above

Reviewer 2 ·

Basic reporting

The authors have addressed the reviewers' comments and concerns in their revised manuscript, but I have a few comments/suggestions for the quality of the paper.
- The main problem in architectures built with homomorphic encryption is the "resource-performance" challenges. In the introduction, it would be better for the quality of the paper to write the motivation subsection and the challenges in these systems more clearly and to continue the solution proposal in articles.
- Detailing how the proposed scheme and decentralization are integrated into the source-performance difficulty in homomorphic encryption would be better for the article's quality.
- It would be better to specify precisely where and how decentralization is used in the specified scheme, both in the motivation and in the content in detail.

Experimental design

- It would be better to make at least a qualitative comparison with other existing architectures.

Validity of the findings

-It would be better for the quality of the paper to determine the computational complexity according to the algorithm specified in section 3.2, to show the complexity specified in sections 5.3 and 5.4 on what basis this conclusion is reached, and to show it qualitatively or quantitatively as a detailed table.

Cite this review as

Reviewer 3 ·

Basic reporting

The authors revise some of the concerns.
- The authors state that the focus of their future research will be on the security aspects of the CC, CA, KC, PC systems. Since this topic can be considered independent of the proposed system, it is appropriate for it to be the subject of future research.
However, in the future, the authors also mention that in their proposed system, analyses will be addressed concerning the fundamental principles of information security such as confidentiality, availability, integrity, authentication, and non-repudiation. Since security is one of the main topics of this system, it is more suitable to include it in this paper.

- Sequence diagrams hold significant importance in representing security protocols and schemes. The diagram in Figure 1 does not show the order of execution, making it difficult to comprehend the process flow in the article. It is evident which operations are being executed from the diagram, but the order in which they are carried out is not clear. Therefore, to present the data flow of the entire scenario in a sequential manner, a sequence diagram is needed.

Experimental design

-

Validity of the findings

-

Cite this review as

---

## Round 0.3 · accepted · Accept

We are happy to inform you that your manuscript has been accepted for publication since the reviewers' comments have been addressed.

·

Basic reporting

Authors updated the paper as per my comments.

Experimental design

As above

Validity of the findings

As above

Additional comments

As above

Reviewer 2 ·

Basic reporting

The authors have addressed my comments and concerns in their revised manuscript.

Experimental design

as above

Validity of the findings

as above

Additional comments

as above

Cite this review as

Reviewer 3 ·

Basic reporting

The authors make the necessary revisions. It is acceptable as it is.

Experimental design

-

Validity of the findings

-

Additional comments

-

Cite this review as